# Space Syntax at Expression of Science on User Flows in Open and Closed Spaces Aimed at Achieving the Sustainable Development Goal: A Review

Paulo Wladinir da Luz Leite [1], Caliane Christie Oliveira de Almeida Silva [1], Leila Dal Moro [1], Brian William Bodah [1,2,3], Giana de Vargas Mores [1], Dirceu Piccinato Junior [1], Amanda Engel [1], M. Santosh [4,5] and Alcindo Neckel [1,*]

1   ATITUS Educação, Passo Fundo 99070-220, Brazil; paulo.da.luz.leite@gmail.com (P.W.d.L.L.);
    caliane.silva@atitus.edu.br (C.C.O.d.A.S.); leila.moro@atitus.edu.br (L.D.M.); bbodah@yvcc.edu (B.W.B.);
    giana.mores@atitus.edu.br (G.d.V.M.); dirceu.piccinato@atitus.edu.br (D.P.J.);
    amandaengel2105@hotmail.com (A.E.)
2   Thaines and Bodah Center for Education and Development, 840 South Meadowlark Lane,
    Othello, WA 99344, USA
3   Workforce Education & Applied Baccalaureate Programs, Yakima Valley College, South 16th Avenue & Nob
    Hill Boulevard, Yakima, WA 98902, USA
4   School of Earth Sciences and Resources, China University of Geosciences Beijing, Beijing 100083, China;
    santosh@cugb.edu.cn
5   Department of Earth Science, University of Adelaide, Adelaide, SA 5005, Australia
*   Correspondence: alcindo.neckel@atitus.edu.br

**Abstract:** Space syntax is utilized to model flows through open and closed built environments, which enables project innovation by rethinking the design of spaces on a global scale with better flow quality. Therefore, studies focused on spatial syntax, when related to comprehensive flows in open and closed spaces, provide a holistic and valuable understanding of the dynamics of user flows in the urban environment from a perspective centered on the United Nation's Sustainable Development Goal (SDG) 11. This goal requires urban planners and designers to use approaches that support future decisions focused on urban design and planning. The object of investigation of this bibliographic study consists of an approach to representing space syntax in open spaces and closed spaces on a global scale. This study aims to analyze the concepts of pedestrian flows (open and closed) by a space syntax-based bibliographical approach on a global scale, demonstrating the capability of improvements in SDG 11 as applied to the architecture of sustainable flows. Methodologically, bibliographic searches were carried out using the Preferred Reporting Items for Systematic Reviews and Meta-Analyses (PRISMA) method in databases linked to Scopus and ScienceDirect, focusing on space syntax in relation to the following terms: "open spaces" and "closed spaces". Using the PRISMA method, 1986 manuscripts dealing with the term "open spaces" and 454 manuscripts dealing with the term "closed spaces" were identified, with the manuscripts most relevant to the topic being selected, followed by a frequency analysis based on a Content Analysis Method (CAM) to identify words with a degree of similarity, related to "spatial syntax: flow in urban environments" and "spatial syntax in closed built systems" in relation to the SDG 11. The results demonstrate the relevance of seven manuscripts in open spaces and four manuscripts in closed spaces related to space syntax. Frequency analysis identified open spaces and produced terms with frequencies such as space = 79 and shape = 46, showing a higher frequency in flows. In closed spaces, the literature has shown that the central term corresponds to space = 79 and flow = 76, making it possible to evaluate flows in circulation areas within the built environment. This study allows a better understanding of flows, highlighting the importance of the urban architecture in the functionality of user flows in sustainable environments, which is capable of contributing to the SDG 11, in the interface of architectural projects on a global scale.

**Keywords:** space syntax; global sustainability; sustainable projects; architectural vision; sustainability management

## 1. Introduction

The built environment requires physical spaces suited to the needs of users (people who travel on pedestrian routes) [1,2]. It is extremely important to have a firm grasp on what routes pedestrians are likely to utilize in order to effectively make appropriate future improvements to the movement and flow of these users. Consequently, an effective urban design considers spatial syntax studies capable of evaluating the quality and comfort of pedestrian flows [2,3]. Therein lies the importance of this bibliographic study, related to the applied use of user displacement dynamics, through space syntax, in the evaluation of open spaces and closed spaces worldwide. The dynamics of pedestrian flows can be understood through the use of space syntax and through the application of the following analytical factors [4–8]: (a) Space analysis, when referring to social interactions represented in a non-linear context; (b) buildings and common spaces between them, as well as the interior of a residence [5–8]; analysis of the visibility graph, using it to study an object, which represents different behaviors [4]; and analysis of the agents based on virtual individuals (called agents) implanted in the virtual environments, where they move around tracing paths and diversified routes so that, at the end of the simulation analysis, the circulation area with greater flow is evident [5–8]. The result of these movements enhances the construction of a synthesized map, where the integration and connectivity flows are displayed. These flows are represented by a color scale, generalized by a flow index which is represented through the colors red (with maximum flow intensity) and blue (with lower flow intensity) [4].

There have been some studies related to the flow of built environments with the use of space syntax for the evaluation of and improvements in the functionality of paths allocated in architectural projects [8–10]. The applicability of space syntax in the area of architecture and urbanism has enhanced its importance. When attributed to the allocation of spaces and environments in the design phase, the use of space syntax affords designers a greater design functionality, in relation to the flow of environments [8,10]. These functional aspects can be applied through the theory of space syntax, created in London in 1970 by Bill Hillier and collaborators from University College London. The book "The social logic of space" was co-authored by Hillier along with Julienne Hanson, and published by the university [11] with the aim of analyzing the functioning of public and private spaces based on quantitative metrics present at different scales of the architectural project. It also presents important aspects of the urban system with universal accessibility focused on natural displacement patterns.

The forms of representation of human movements in the built environment through the analysis of space syntax are understood as a set of movements related to individuals, with the aim of providing the theory of natural movement [12]. This concept can be understood through the ability of people to move along paths in the built environment [12,13]. The resulting pedestrian flow through a design can be adequately predicted utilizing space syntax, thus providing information on the daily needs of individuals in walking urban routes or built environments [9,13].

Easy mobility within a given area or space enables individuals to engage in increasing social interaction in these spaces intended for pedestrian movement [14,15]. By maintaining consistently active pedestrian movement, a space can be designed that allows for rapid transit and ease of transit with safety and attractiveness to passers-by [14,15]. Importance is placed on effective, rapid, and easy pedestrian movement, which are considered important in maintaining the quality of life of users and in the better organization of public spaces [16,17].

Yang et al. [18] highlights the urban fabric as a way of representing the configuration of the urban design. It involves different pedestrian routes that make up the structure

of the environment without neglecting the consideration of both positive and negative points aimed at the accessibility of individuals. This is applied to both open spaces (public walkways) and closed/restricted spaces (internally built systems). The need for this study, using the theory of space syntax in a bibliographical way, is justified as the association between the design of an environment and the efficiency with which pedestrians flow through it, is well documented [19]. Space syntax has stood out for predicting results based on the natural movement of people for several decades [20]. The circulation of users reflects on the form of urban mobility in relation to the different routes chosen and utilized by users [20,21].

The inclusive circulation of users in cities was guaranteed at the United Nations General Assembly in 2015 by the goal called "Sustainable Cities and Communities" by SDG 11 [22]: the importance of mobility focused on pedestrian movements with safety, resilience and sustainability is highlighted, with rights guaranteed by efficient urban public policies, which will contribute to improving the quality of life of the population on a global scale [22].

When dealing with the inclusive movement of the population in certain urban centers, Yıldırım and Çelik [23] emphasize that sustainability related to pedestrian displacement and the perceived relations of safety and freedom of users, when carrying out their movements, makes the routes in the built environment in pleasant places. According to Yıldırım and Çelik [23], it is up to the architect and urban planner to apply the space syntax methodology to design urban spaces aimed at pedestrian movement that are more suitable for sustainability in favor of sustainable urban security and resilience. For Datola [24], Tang and Chen [25], urban resilience aimed at human displacement consists of the ability to adapt and transform cities quickly and efficiently to meet new possibilities, such as obtaining population movements with universal accessibility.

Achieving sustainability suggested by the SDG 11 [22], when linked to achieving urban pedestrian mobility, with safety and resilience [25,26], requires more studies of space syntax at a global level to comprehend real problems, which are generally worsened by excessive user flows during the process of commuting in cities. These displacements, when represented by space syntax, allow designers, in the project design phase, to create projects for open and closed spaces (built environments) and design more attractive environments for pedestrian movements [27,28]. According to the SDG 11 [22], we need to achieve more inclusive cities, with greater safety and resilience to pedestrian movement patterns, aimed at sustainability at a global level.

The general objective of this study is to analyze the concepts of pedestrian flows (open and closed) by a space syntax-based bibliographical approach on a global scale, demonstrating the capability of improvements in Sustainable Development Goals (SDGs) as applied to the architecture of sustainable flows. In order to accomplish this, a bibliographical analysis of the following perceptions was performed: space syntax, and flow in urban environments and space syntax in closed built systems. The space syntax in open spaces deals with the analysis carried out in paths through a direct analysis of the urban project, aimed at user circulation in virtual spaces [23,29–34]. The space syntax in closed spaces deals with the analysis of the flow of users in the internal environments of buildings, with the purpose of a direct analysis of the efficiency of flows in the architectural design of a given space [5–8].

Tanaka et al. [7], Yildirim and Çelik [23], and Soltani et al. [34], highlight the need to carry out further studies utilizing space syntax globally, since there is serious lack of dissemination of knowledge of the techniques of application of space syntax currently. According to Dawodu et al. [35], sustainability can be incorporated into this debate from the perspective of architectural design focused on sustainability, reflecting on the ideas of optimizing spaces for more adequate user flows. This situation aims to make the process from design to construction economically viable and provide comfort and well-being to users [35].

## 2. Materials and Methods

This mixed study included systematic bibliographic reviews, constructed from the Preferred Reporting Items for Systematic Reviews and Meta-Analyses (PRISMA) [36,37], through searches in databases linked to Scopus and ScienceDirect between 2019 and 2022 related to the topic of space syntax in open or closed spaces, with the development of case studies. The use of the PRISMA methodology [36,37] considered analyses of manuscripts published in the English language, with a total of 1986 manuscripts that deal with the term "open spaces" and 454 manuscripts with "closed spaces", that is, 2440 sources of databases. Through PRISMA filtering, with parameters for choosing manuscripts with high reliability [37,38], the relevance of seven manuscripts with open spaces and four manuscripts with closed spaces was obtained, when related to the application of space syntax (Figure 1).

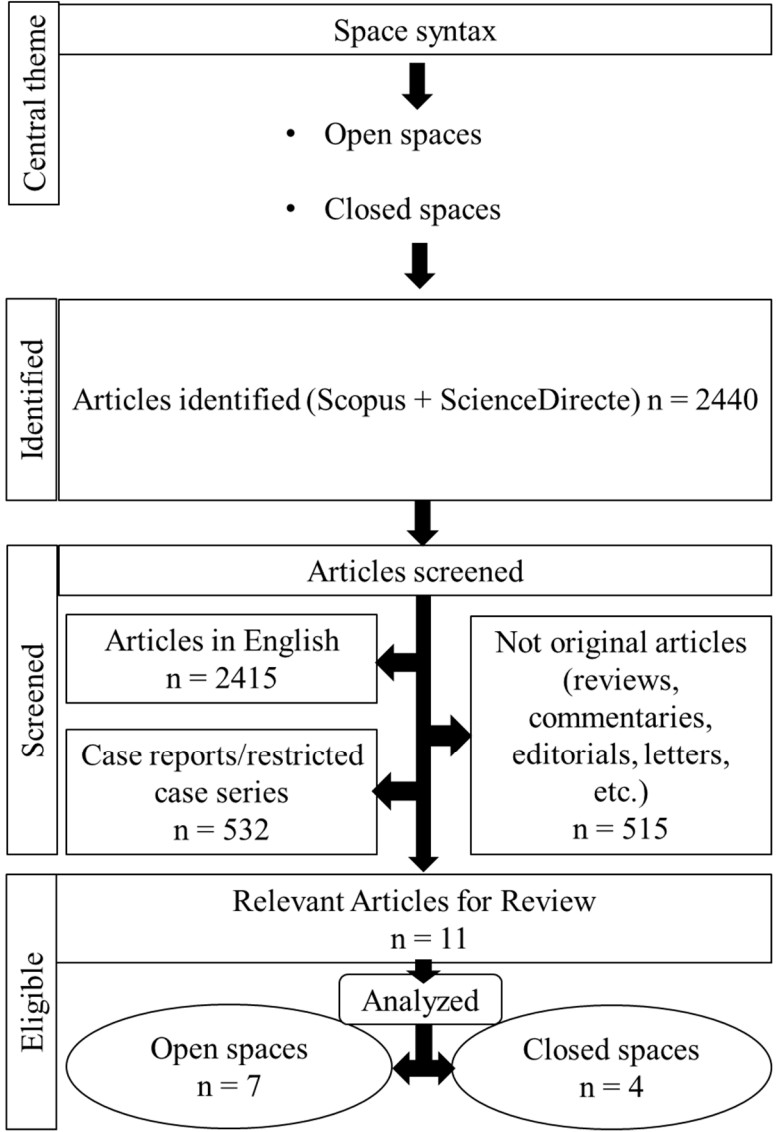

**Figure 1.** Representation of the Preferred Reporting Items for Systematic Reviews and Meta-Analyses (PRISMA) model [36–38] applied to open spaces and closed spaces related to spatial syntax.

After selecting open and closed space syntax manuscripts selected from the PRISMA method, frequency analyzes were carried out, using the Content Analysis Method (CAM) [29–31,39,40], for identification of the most used terms in the selected manuscripts through the degree of similarity by frequency between the space syntax studies identified

in the literature. CAM is a technique that uses bibliographic material to be analyzed and classified by themes focused on the classification detected by the observing researcher in relation to the understanding of discourse [41,42], between open and closed spaces related to spatial syntax. Considering the application of CAM in the selected manuscripts [36–38], the verification of terms that are repeated in these manuscripts with greater frequency is presented [39,40].

## 3. Results and Discussion

### 3.1. Space Syntax: Flow in Urban Environments

The diverse natural forms of pedestrian displacement have enhanced the analysis of the studies recorded in the Scopus and ScienceDirect database. A total of 1986 existing metadata files were returned when utilizing the search term (based on space syntax in open spaces). However, only seven of the returned files were characterized as being studies related to the topic addressed in this study [23,29–34], the utilization of space syntax in the evaluation of flows in environments with open paths aimed at pedestrian activities (Figure 2).

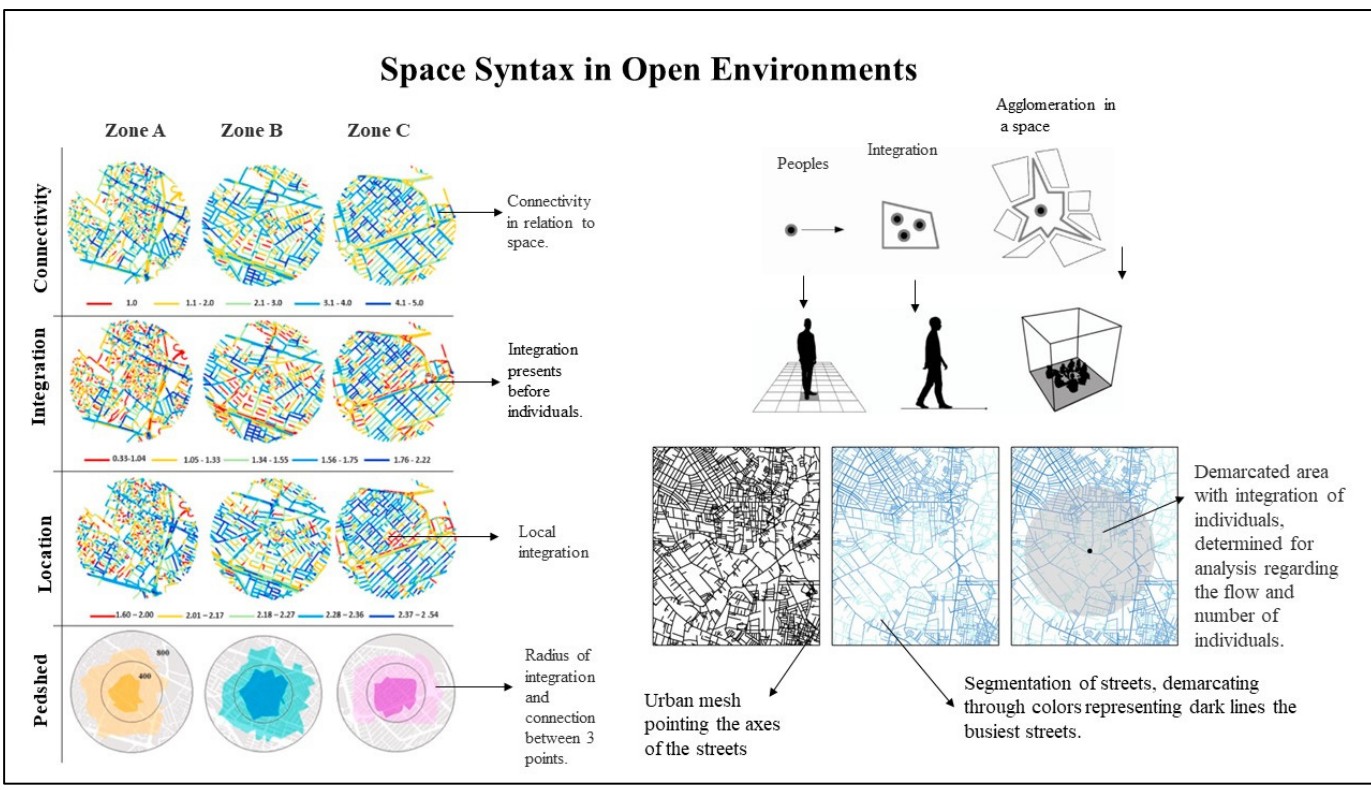

**Figure 2.** Presents the concepts of the studies by Yildirim and Çelik [23], Koohsari et al. [29], Tannous et al. [30], El-Darwish [31], Garau and Annunziata [32], Gharaibeh et al. [33], and Soltani et al. [34]. These studies utilized space syntax in open environments along with CAM to reveal the following terms of greatest frequency: pedestrian trips; mobility; urban mesh; space; form; and quality of life.

Yıldırım and Çelik [23] applied space syntax for analyzing the behavior of pedestrians in certain areas in the Besiktas region of Istanbul, Turkey. The authors sought solutions for future projects to enhance the urban fabric of the city based on an analysis of pedestrian displacements and daily paths taken by virtual individuals. The importance of pedestrian displacements in public spaces is highlighted, where certain types of infrastructure can lead to changes in the paths pedestrians utilize in addition to impacting the best location of urban roads [23].

Koohsari et al. [29], utilized space syntax to analyze the natural movements of individuals and showed the importance of pedestrian displacement in certain urban spaces by enhancing pedestrian circulation in relation to that displacement. Such pedestrian movements, when successful, can lead to improvements in the quality of life of users [29]. In view of this, analyzing walks as a relevant factor for urban mobility, using both space syntax and the Dephmath (Version 50) software, requires further investigations in relation to representation by simulation in the urban displacement of individuals [29]. Dephmath is a software used in the areas of urban design, architecture, urbanism, transport and geographic sciences, among others. The use of these two technologies together enhances improvements to be attributed in future urban projects, capable of solving possible problems that users face when moving through the built environment as pedestrians.

Tannous et al. [30], when discussing accessibility in the context of green areas in a metropolitan region of Doha, Qatar, used space syntax to evaluate the spatial configuration of public sidewalks through the use of flow analysis over displacement of people. The importance of green spaces aimed at accessibility in open environments was highlighted in this study. The authors also emphasized the importance of spaces that address challenges related to accessibility, highlighting the importance of space syntax for the delineation of routes with greater pedestrian integration [30].

On the other hand, El-Darwish [31] focused on an analysis of social interaction within a university space, examining the efficiency of pedestrian movements on certain routes. The increase in integrality, interaction and quality of life stands out when related to the importance of displacement carried out in open spaces. Through the results of El-Darwish [31], it is possible to model pedestrian routes with a greater and more efficient flow of users, enabling the design of new spaces for walking, while at the same time enhancing the comfort and quality of life of users.

Garau and Annunziata [32] applied space syntax analyses to open public spaces in the Villanova district, in Cagliari, Italy. They sought a connection between the forms of environmental displacement through the application of analytical methods defining the quantitative characteristics of urban space and pedestrian movements. These pedestrian movements have a potential relation to the quality of life of users, through social activities [32]. The results of Garau and Annunziata [32] demonstrate that by studying the displacement relations between users, it is possible to quantify pedestrian circulation movements and guarantee an understanding of the integrality of the circulation spaces among users.

Gharaibeh et al. [33] used space syntax to analyze urban environments through the use of new tools capable of aggregating quantitative pedestrian mobility indices, aimed at universal accessibility in relation to the public transport system. Their results demonstrate the importance of demarcating analytical points (pedestrian paths) in addition to the efficiency of using different software (Gravity Model (2020 version (xGRAV20)); Weighted Overlay (version 10.3)) along with space syntax to carry out new research. Furthermore, this study demonstrated space syntax's applicability when determining how best to encourage the public's use of public transport (buse and trains, among others) [33].

Soltani et al. [34], employed space syntax to carry out urban analyses using a road radius of 400 to 800 m in order to verify the daily pedestrian displacement of individuals within a given path. The study highlighted the influence open space has on pedestrian movements within the urban fabric. Their findings were consistent with those of Garau and Annunziata [34], showing that pedestrian movements cause greater interaction between and among users.

Based on the five categories of analysis determined a priori based on the literature [39,40] in relation to open environments, the frequency was presented by enumeration. In this context, the following frequencies of terms by numerical repetitions were considered (Table 1): mobility = 16; quality of life = 21; urban mesh = 30; shape = 46; and space = 79. These categories are interconnected and play an important role in the development of cities. Furthermore, adequate urban planning becomes fundamental to create sustainable projects

for environments that are more accessible to population displacement [31]. Considering the highest number (frequency), the intelligent use of space when reaching a frequency of 79 becomes compatible with mobility, quality of life and sustainability of the built environment. In relation to the creation of adequate infrastructure, this ensures universal accessibility for pedestrians [32].

**Table 1.** Space syntax in open environments identified by frequency from CAM.

| Authors | Objective | Result/Conclusion | Positivity of the Analysis |
|---|---|---|---|
| Yıldırım and Çelik [23] | Verify modeled movements of pedestrians throughout the urban fabric. | Better understand how pedestrian movements are governed by the constructed urban fabric. | Pedestrian movement can be influenced by the design of the urban fabric. |
| Koohsari et al. [29] | Emphasize the importance of pedestrian movements in relation to the mobility of space due to the layout of the urban fabric. | The need for studies of movements in the urban fabric, in conjunction with urban architects and geographers. | Projections of pedestrian movements in the urban fabric need improvement through the creation of new software. |
| Tannous et al. [30] | Identify the importance of pedestrian movements in the urban fabric. | The analysis of the urban fabric becomes fundamental in the pursuit of accessibility. | Paths taken by individuals in the urban fabric have the ability to improve the quality of life of citizens. |
| El-Darwish [31] | Diagnose types of pedestrian movements in the urban fabric, with actions that contribute to the quality of life of users. | Paths taken by individuals in residential areas need to improve the quality of life of users through alterations in urban infrastructure. | The way of analyzing mobility contributes to improving overall urban infrastructure in order to seek a higher quality of life for individual users. |
| Garau and Annunziata [32] | Analyze the pedestrian movements of people in the urban fabric. | The study shows the importance of the quality of the space's layout, which increases the users' quality of life. | Software can be utilized to better design pedestrian movement through the urban fabric. |
| Gharaibeh et al. [33] | Analyze accessibility in pedestrian movements in order to demarcate transport areas in the urban fabric. | Through the analysis of the urban fabric, new routes can be designed in this space, in addition to the implementation of new ways to encourage the use of public transport. | The analysis of the urban fabric enhances future research in search of computational programs capable of contributing to the overall design of the urban built environment. |
| Soltani et al. [34] | Analyze the urban fabric and the daily pedestrian movements of individuals as they move through the urban fabric. | The pedestrian movements of individuals support the interaction of people in the urban fabric. | Adequate mobility in the urban fabric provides improvements in the road network in order to designate the best placement of new roads and pedestrian paths. |

Source: Based on Yildirim and Çelik [23], Koohsari et al. [29], Tannous et al. [30], El-Darwish [31], Garau and Annunziata [32], Gharaibeh et al. [33], and Soltani et al. [34].

By observing the factors integrated into a space, after analyzing the studies of Yildirim and Çelik [23], Koohsari et al. [29], Tannous et al. [30], El-Darwish [31], Garau and Annunziata [32], Gharaibeh et al. [33], and Soltani et al. [34], the topics are highlighted in relation to pedestrian movements when analyzed in different highlighted perceptions (Figure 3). These include the following: Mobility (1) represents a form of displacement and a path that users utilize/take in a space [43]; Urban mesh (2) is defined by an occupied area designated as within an urban perimeter, represented by blocks with lots and streets, and composed of urban solids and voids [44]; Space (3) is defined as an area that encompasses a group of dwellings, horizontal and vertical buildings, for industrial, leisure, social or cultural use [45]; Form (4) is the way in which an object is found in a given space in relation to the concept and design guidelines, based on the conception presented by an architectural project [46]; Quality of life (5) is a measure of perception an individual has regarding their overall quality and way of life [47,48].

The terms pedestrian movement, mobility, urban fabric, space, form and quality of life were identified and presented in studies by Yildirim and Çelik [23], Koohsari et al. [29], Tannous et al. [30], El-Darwish [31], Garau and Annunziata [32], Gharaibeh et al. [33], and Soltani et al. [34]. Measuring and modeling pedestrian displacement is the main reason for carrying out studies utilizing space syntax in open environments (spaces not enclosed within buildings). This technique makes it possible to predict the best layout of walking paths in an analysis of the space to be developed in the urban fabric, in order to increase quality of life for all users.

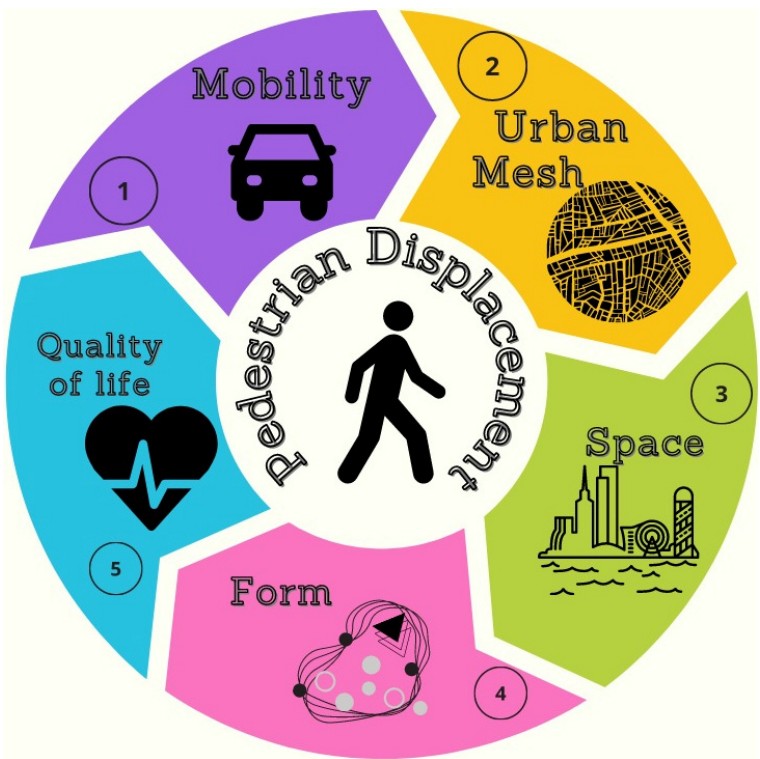

**Figure 3.** Representation of frequencies in urban environments with open pedestrian flows. Modified from the following sources: Yildirim and Çelik [23], Koohsari et al. [29], Tannous et al. [30], El-Darwish [31], Garau and Annunziata [32], Gharaibeh et al. [33], and Soltani et al. [34].

Despite the importance of studying open spaces (streets, roads and pedestrian paths) through space syntax, Tanaka et al. [7] and Zerouati [8] have shown that there is also a need to apply space syntax to development plans of closed environments (areas within buildings).

*3.2. Space Syntax in Closed Built Systems*

A search for patterns of pedestrian movement in the built environment gave rise to studies, found in Scopus and ScienceDirect, identifying a total of 454 metadata files, each of which included the following search terms: space syntax and indoor environments (space syntax; indoor environments). Among these, four files qualified in this screening, as they are studies that address flows in internal built systems, considering the potential of pedestrian flow using space syntax [5–8] (Figure 4).

Alitajer and Nojoumi [5] questioned residential privacy in their study, which encompasses the pedestrian movements of individuals within the space configuration of traditional and modern houses in the city of Hamedan, Iran. Using space syntax as an analytical form aimed at verifying movements, quantitative comparisons were carried out using Dephmath software [5]. The influence on the projection of new buildings in relation to the integration between the existing built environment was necessary in order to predict the displacement of individuals between and within the two (new construction and existing

infrastructure) [5]. The results recommend the analysis of space syntax in the initial phase of the architectural project. This allows for possible design changes in order to improve the functionality of the building's flows [5].

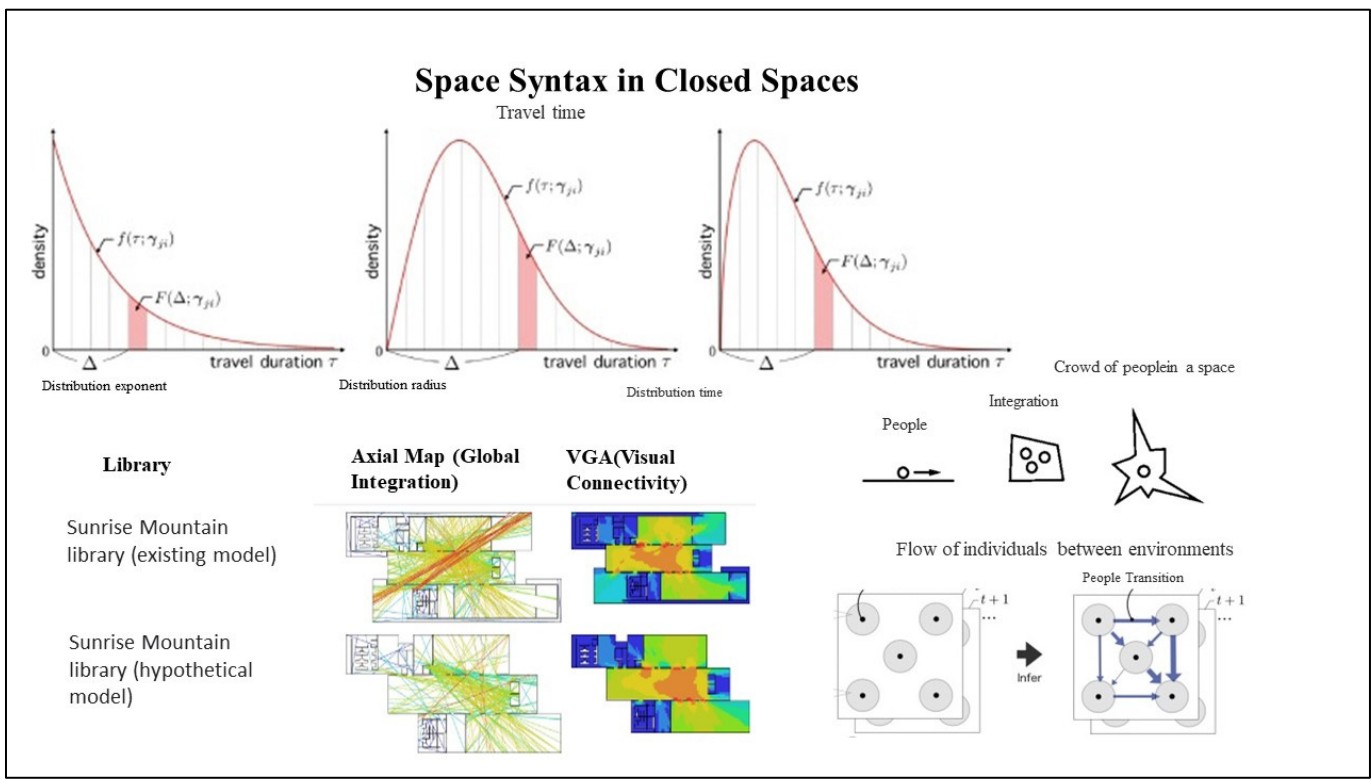

**Figure 4.** Representative conceptions of space syntax in a closed-flow environment. Source: Alitajer and Nojoumi [5], Askarizad and Safari [6], Tanaka et al. [7], and Zerouati [8].

According to Askarizad and Safari [6], when analyzing the function of semi-open spaces in an effort to understand the behavior of individuals in relation to pedestrian movements within a public library of Sunrise Mountain and Desert Broom Libraries (Arizona, USA), it is possible to obtain data capable of aggregating projections to be applied in future public buildings.

From this, space syntax was used as an analytical method [6], and the importance of pedestrian displacements in built systems was highlighted, where it enabled the identification of perceptions of insecurity in users who moved in a building [6]. Space syntax, in addition to identifying the areas of greatest flow, made it possible to adjust security in periods of intense user movement.

According to Tanaka et al. [7], using quantitative statistical methods and models of people flow spread to identify possible displacement data within a built system, through collection points in order, makes it possible to analyze the flow of people in relation to the time of entry and exit from the building. This study utilized a shopping mall with exhibition halls. The use of space syntax for data analysis in relation to the enclosed pedestrian flows within this building made it possible to understand the displacements necessary to carry out expansion projects [7].

Zerouati [8] used the analysis of space syntax in historical buildings, at an archaeological site located in Setif, Algeria. The objective of their study was to investigate the likely quality of life in relation to the flow and pedestrian displacement that took place within this built environment. Their results indicate the relationship between the internal spaces of the historic building and the quality of life those pedestrians would have likely experienced based on projected broad flows and internal displacements [8].

Based on the analysis represented in the literature [49], it was possible to carry out frequency analysis, through the use of CAM (which has five stages), in order to identify the following main points: (1) problem; (2) possibilities; (3) explore data; (4) test fundamentals and (5) present the solution. Therefore, the use of CAM is presented and attributed to data researched in the literature [49].

By assuming the frequency analysis by CAM from Alitajer and Nojoumi [5], Askarizad and Safari [6], Tanaka et al. [7] and Zerouati [8], the following terms resulting from the analysis were identified: space, movements, environment, flow, form and circulation (Table 2). It was possible to highlight, from the terms identified by frequency, the following conceptions of space syntax in studies in built environments with closed characteristics: space is the built environment, where studies of flow and circulation movements take place by space syntax in the initial phase of conceiving the form of the architectural project.

**Table 2.** Closed space: syntax identified by frequency from CAM.

| Authors | Objective | Result/Conclusion | Positivity of the Analysis |
|---|---|---|---|
| Alitajer and Nojoumi [5] | Analyze the movements of individuals in internal spaces in traditional and modern buildings in Hamedan using space syntax. | The influence on the design of a building is favored by the integration of the environment with the circulation of people. | The use of auxiliary software at the time of the project facilitates the analysis of the form, with the understanding of the movements of the individuals in the projected environment, contributing to the improvement of the conceived space. |
| Askarizad and Safari [6] | Understand people's behavior and their movements inside the built space. | The study of flow in built space helps urban architects in the initial phase of the project. | The importance of open space, with analysis of the movement of people causing greater security for users. |
| Tanaka et al. [7] | Check with a statistical method the disposition, time of entry and exit, form and flows of people in the built space. | Analyses with data in space subdivided into points in the environment with the greatest flow of movements, optimizing flow within the building. | The disposition and time in relation to the flow of individuals. Such movements can be understood in relation to the number of people in the projected space. |
| Zerouati [8] | Analyze the quality of life in a built space. | The form of flow hierarchy stands out in the different points of view of the designer. | The way of analyzing the flow of people varies according to the environment in relation to public or private spaces. |

Source: Alitajer and Nojoumi [5], Askarizad and Safari [6], Tanaka et al. [7], and Zerouati [8].

The seven categories of indoor analysis were presented by repetitive frequency numbers [49]. Such categories were used a priori based on the literature and the frequency established in the final compiled data [39,40]. Following are the terms in relation to their frequencies: circulation = 14; movements = 31; shape = 46; environment = 52; flow = 76; space = 79. It is important to reiterate that the most latent points (space and flow) are fundamental for building more efficient, inclusive and sustainable cities [50,51].

The space configuration of the city determines how people move, interact and use the different spaces available [6]. Proper planning for the implementation of projects in urban space can influence flows, facilitating mobility which is capable of contributing to a more functional and sustainable city [5].

When analyzing the different aspects that integrate the internal environments of a building and, incorporating the studies of Alitajer and Nojoumi [5], Askarizad and Safari [6], Tanaka et al. [7] and Zerouati [8], the following points were noted in relation to the internal movement of pedestrians within the space (Figure 5): Movements (1) are considered pedestrian movements over short distances and take place along different paths and routes [52]; the Environment (2) is a space formed to relate individuals to their social environment, conceived in either a material or immaterial way [53]; Flow (3) is a form of movement related to the physical state of individuals that are in the same

place and trace different paths, causing varied flows [7]; Form (4) is the way in which an object interacts in space, in relation to its design concepts and guidelines, based on the architectural concept [46]; and Circulation (5) includes the possibility of movement in relation to individuals being able to move around in the environment [54].

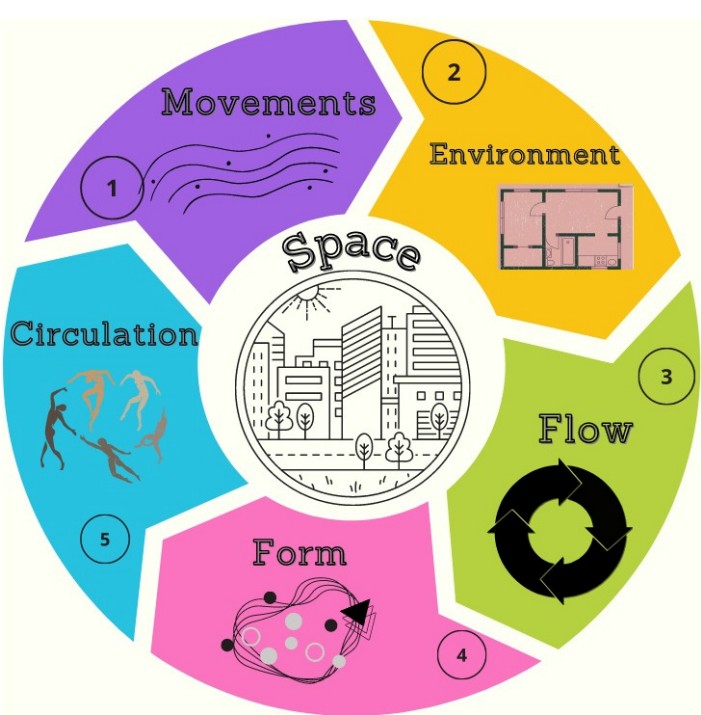

**Figure 5.** Perception of frequency in relation to the terms of space, movements, environment, flow, form, circulation and movements in relation to the space syntax of internal closed built spaces. Modified after the following sources: Alitajer and Nojoumi [5], Askarizad and Safari [6], Tanaka et al. [7], and Zerouati [8].

Alitajer and Nojoumi [5], Askarizad and Safari [6], Tanaka et al. [7] and Zerouati [8] emphasize the need to carry out studies utilizing space syntax worldwide applied to flows in built internal spaces. This study focused on the area of architecture and urbanism, but did not bibliographically identify other studies highlighting internally built spaces with applications of space syntax in the Scopus and ScienceDirect databases. In summary, other research recognizes the importance of using other bibliographic databases with Scopus and ScienceDirect, as they are aimed at understanding pedestrian movements and highlighting the central idea of relevance when attributed on a daily basis to flows occurring on urban roads [54,55].

According to Yiannakoulias and Scott [55] and He et al. [56], the concept of greater relevance in pedestrian displacement comes closer to guaranteeing pedestrian safety by public authorities, as fear of accidents, marginalities, lack of lighting, conservation and maintenance generate certain perceptions for users. Therefore, it is necessary to constantly evaluate pedestrian paths in order to stimulate public policies capable of subsidizing urban architects in an effort to design and create urban projects that will contribute to the quality of life of users when moving around in the environment [56–58].

Another central theme in this study was the term "Space", identified in Alitajer and Nojoumi [5], Askarizad and Safari [6], Tanaka et al. [7] and Zerouati [8] as considering the evaluation of flows in built environments through space syntax. According to Tanaka et al. [7], the assessment of flow space symbolizes a concern that designers acquire about the possibility of improving architectural projects in a sustainable way. When a building is constructed, the identification of the quality of pedestrian flow (both internal and external) is attributed to the efficiency of existing points of access. Enabling alternatives will improve

the mobility of users throughout a building based on changes in the use of architectural compartments of functionalities established in the design phase of the project [59,60].

## 4. User Flows in Open and Closed Spaces, with the UN SDGs

The advancement of initiatives on Sustainable Development Goals (SDGs) have been growing on a global scale. However, these initiatives depend on the understanding of and interaction between local authorities, themes, benefits and actions in a positive and effective way. The scope of displacement spaces that enhance greater accessibility of user flows in open and closed environments depend on these [61]. The SDGs require advances in tackling problems arising from the implementation of future architectural projects capable of interconnecting private and public areas of the urban environment [62].

There is a connection between actions carried out and the disclosure of those actions. The global news on different topics with a focus on the 2030 Agenda, with 2.5% contemplating the goals in different topics, attribute the need for discussions that will improve urban flows on a global scale [63,64]. Consequently, both space syntax studies in environments with closed flows [5–8] and in urban spaces with open pedestrian flows (streets and public spaces) [23,29–34] emphasize the importance of applying improvements to architectural projects aimed at the sustainability and effectivity of pedestrian flows in the urban environment. The flow of users in open and closed spaces is not widely mentioned in the SDGs. Additionally, many objectives aimed at the sustainability of the built environment are related to the flows of pedestrian users [65,66].

One of the objectives most related to the flow of users in open spaces is the SDG 11 (Sustainable Cities and Communities) [67]. The SDG 11 targets making cities and human settlements inclusive, safe, resilient and sustainable. This includes a number of specific targets that may be relevant to user flow, such as creating safe and accessible public spaces and reducing accidents related to traffic, with an emphasis on greater safety in the pedestrian movements of users. Also, it includes increasing planning capacity with more sustainable urban management [61].

To promote the safe and sustainable flow of users in open and closed spaces, it is important that governments and communities work together to achieve these goals, creating policies and practices that promote the safety, accessibility and sustainability of cities and communities [61,66,67]. Urban displacement plays an important role in global development trajectories worldwide, as recognized by the 2030 Agenda [67].

In the dimension of an architectural project, sustainability aimed at displacement in open and closed flows starts from the urban analysis of the built system. In the sense of valuing sustainable use with integrated and detailed planning of the urban process of pedestrian flow, the architectural product is the result of multidisciplinary actions in compliance with norms and legislation [67]. These principles share support for sustainable architecture projects that improve the flow of users in both open and closed spaces. Raising awareness, guiding and clarifying the functioning and benefits of these spaces, and studies utilizing space syntax all allow for a marked increase in sustainable architectural design [68].

Carrying out additional space syntax studies (Figures 3 and 5) enables a better understanding of and the possibility of proposing more comprehensive structures that address user flows in open and closed spaces. Adequate planning to alleviate future issues around user flows, resolving them while still in the design phase, is the goal here. This increasingly enables the existence of an environment with greater sustainability applied to differing potential pedestrian routes [69,70].

*Validation of Bibliographies Using the PRISMA Method and Frequency Analysis Using CAM Integrated with the SDG 11*

Manuscripts selected by the PRISMA method [36–38] and applied to open spaces [23,29–34] and closed spaces [5–8], based on applicability of space syntax when analyzed by frequency and using CAM [39,40], demonstrated different frequencies. This difference attributed by CAM generated homogeneous and heterogeneous terms for the application of space syntax in

open spaces (mobility = 16; quality of life = 21; urban mesh = 30; shape = 46; and space = 79) and closed spaces (circulation = 14; movements = 31; shape = 46; environment = 52; flow = 76; and space = 79). The relationship between the homogenous terms of space syntax analyzed is enhanced by studies related to the scope of form and space, based on open or closed urban environments [5–8,23,29–34].

The term "form", within the scope of the applicability of space syntax, consists of open spaces related to urban form, when considering the connections of paths in the urban fabric, where the dynamics of pedestrian movements occur [29,33,71,72]. In closed spaces, the form is established by the built structure, represented in areas where people circulate [7,8,73,74]. In this context, space syntax, when dealing with form in open and closed spaces within the scope of sustainability when considering the SDG 11 [67], is based on cities with more inclusive travel spaces, with the absence of barriers or obstacles, capable of guaranteeing universal accessibility on routes intended for the movement of people [62,75,76]. This guarantee of the SDG 11 allocation in relation to improvements in travel paths in open and closed spaces must be considered by designers in the design phase of an urban or architectural project [77–79]. The relationship of space in space syntax studies applied to studies of open [23,29–34] and closed [5–8] spaces presents a general term, in that follow different forms of movement on urban routes and in built spaces intended for the movement of people.

In this context, the relationship between the special syntax of open spaces and closed spaces aimed at sustainability, in relation to the studies analyzed [5–8,23,29–34], discuss the dynamics of user movements through likely pedestrian movement. Appropriate routes are necessary for better travel, which requires investment in universal accessibility projects in urban environments in order to facilitate different forms of pedestrian movement for users in both open spaces and closed spaces (Figure 6).

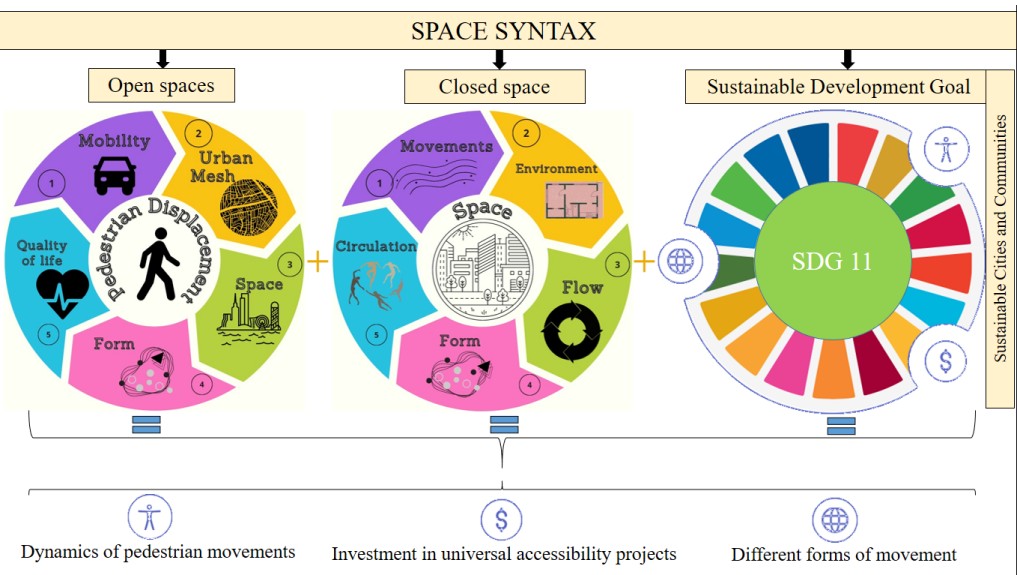

**Figure 6.** The spatial syntax dimension of pedestrian movements in open spaces and closed spaces within the scope of SDG 11.

Tannous et al. [30] treat the term as space as a set of different scales within the urban environment. Monokrousou and Yiannopoulou [80] highlight that the term space, in the context of sustainability in urban environments, represents a future challenge for urban designers—being able to design routes that guarantee the most effective movements of people when focused on quality m—making it possible to limit the private and excessive use of motorized vehicles. In this study, when relating the terms form and space, the terms applied to space syntax in open spaces stand out, such as: mobility, quality of life and urban fabric [23,29–34]; and closed built spaces, such as circulation, movements, environment and

flow [5–8]. Thus, it was highlighted that the form allocated within a given space enables relationships of mobility, quality of life, urban fabric, circulation, movements, environment and flow (when observed in space syntax case studies [5–8,23,29–34], analyzed by frequency by CAM, between 2019 and 2022), which are part of the human relations of displacements between different studies on a global scale.

According to Esposito et al. [81], related space syntax studies in environments with pervasive flows in open and closed spaces provide a holistic and valuable understanding of the dynamics of user flows. Therefore, the focused perspective of sustainable urban development objectives needs to be adopted by urban planners and designers in order to support urban design and planning decisions, based on analyses applied to the urban physical environment [81]. In this context [81], the field of cognitive sciences globally addresses space syntax techniques aimed at understanding the relationships of cognitive spatial agents directed in approaches related to their characteristics in different spatial contexts in urban environments.

## 5. Conclusions

The analysis of the conceptions of pedestrian flows examined utilizing space syntax, identified in the literature, allowed for a better understanding of pedestrian flow within both open and closed spaces. This study highlighted the importance that some international architects attribute to the evaluation of flows through the simulation of pedestrian flows in the environment, using space syntax. The space syntax analysis method could become widespread by architects and urban planners on a global scale, which would enable a better assessment of pedestrian flows through the built environment in the design phase of urban and architectural projects. The authors suggest a greater use of space syntax in studies conducted world-wide that focus on a relationship with UN SDGs in both open and closed spaces in order to expand what is available in the literature. In addition, the use of space syntax enables greater innovation aimed at quantifying information by analyzing flows in geographic space. It becomes impossible to innovate for the greater and more efficient usage of pedestrian space without being able to know or predict likely patterns of pedestrian movement. Models such as space syntax greatly assist in these simulations.

The literature consulted the analysis of open spaces and yielded terms with the frequencies space = 79 and shape = 46, presenting a higher frequency in the flows. This attributed the idea of central preference, when focused on the evaluation of urban roads, in the favor of the quality of life of users. In closed spaces, the literature has shown that the central term corresponds to space = 79 and flow = 76, making it possible to evaluate flows in circulation areas within the built environment. Space syntax is capable of solving problems and enabling improvements that can be made during the design phase of an architectural project.

This study reinforces the importance of applying space syntax in open and closed environments on a global scale. Therefore, it is necessary to highlight the few practical observations of spatial syntax found in the literature, focused on open spaces and closed spaces and sustainability of urban environments.

Regarding future research, we suggest that teams examine the application of space syntax based on the literature presented in this study, in addition to the use of other databases. Such research should consider the constructed environment and public use aimed at the displacement of users. This study provides a theoretical understanding of the applicability of studies utilizing space syntax world-wide, in addition to contributing to the unique academic value of applying space syntax analyses in future project design. By addressing likely issues in the design phase, pedestrian flow problems can be eliminated before they even occur.

For future studies, it is necessary to investigate space syntax, addressing other bibliographic sources associated with social interactions, behavior patterns, readability, cognitive maps, a sense of belonging, mobility patterns, spatial cognition, security, orientation and

privacy protection that used space syntax to expand its applicability in SDG 11, in databases published between 2020 and 2023.

**Author Contributions:** Conceptualization, G.d.V.M., L.D.M. and P.W.d.L.L.; data curation, A.N.; formal analysis, C.C.O.d.A.S. and A.N.; funding acquisition, B.W.B.; investigation, P.W.d.L.L.; project administration, C.C.O.d.A.S. and A.N.; supervision, D.P.J. and G.d.V.M.; visualization, A.N.; writing—original draft preparation, A.E. and A.N.; writing—review and editing, M.S. and A.N. All authors have read and agreed to the published version of the manuscript.

**Funding:** This research received no external funding.

**Institutional Review Board Statement:** Not applicable.

**Informed Consent Statement:** Informed consent was obtained from all subjects involved in the study.

**Data Availability Statement:** Data are contained within the article.

**Acknowledgments:** The authors are grateful to the Air Centre for supporting this research and the National Council for Scientific and Technological Development (CNPq) for the research productivity grant in Brazil. The authors also extend their thanks to the Center for Studies and Research on Urban Mobility (NEPMOUR + S/ATITUS), and Fundação Meridional, Brazil.

**Conflicts of Interest:** The authors declare no conflicts of interest.

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
