# Peer review of "Space Syntax at Expression of Science on User Flows in Open and Closed Spaces Aimed at Achieving the Sustainable Development Goal: A Review"

_2673-8945, doi:10.3390/architecture4010011_

Round 1

Reviewer 1 Report

Comments and Suggestions for Authors

This paper delves into the application of space syntax to model pedestrian flows in both open and closed built environments on a global scale. Employing bibliographic methods and the PRISMA technique, the study meticulously analyzes relevant manuscripts, with a specific focus on the intricate concepts of pedestrian flows and their correlation to Sustainable Development Goal (SDG) 11. Emphasizing the paramount role of deliberate planning in urban design, the paper underscores the predictive capabilities of space syntax in crafting environments conducive to pedestrian movement. The conclusion accentuates the significance of space syntax in comprehending and enhancing global pedestrian flows, urging for its broader integration in forthcoming projects. Overall, the paper maintains a well-structured logical flow, ensuring ease of comprehension and aided by elucidative tables.

While the cited literature aligns coherently with the paper's theme and exhibits adequacy in both quantity and quality, the authors can further fortify their work by considering insights from the following publication:

Esposito, D., Santoro, S., & Camarda, D. (2020). "Agent-based Analysis of Urban Spaces Using Space Syntax and Spatial Cognition Approaches: A Case Study in Bari, Italy." Sustainability, 12(11), 4625.

This source offers a holistic understanding of how users experience and navigate urban spaces, providing valuable insights into the dynamics of user flows. The user-centric perspective presented in this paper contributes to a more comprehensive grasp of user flows in open and closed spaces, aligning seamlessly with the objectives of sustainable urban development.

These considerations are expected to enhance the paper, offering a valuable perspective for further research on the topic and making it more suitable for publication.

Comments on the Quality of English Language

Fine

Author Response

Manuscript ID: architecture-2633507

Space Syntax at expression of science on user flows in open and closed spaces aimed at achieving SDGs: A review

 Dear Editor

Thank you for taking the time to manage this editorial process and for sending us review comments of revision.

We have revised our paper accommodating all the comments. We provide below the point by point reply to the comments.

We declare that this paper has not been published previously and it is not under consideration for publication elsewhere. The manuscript does not involve research on humans and animals. The authors also declare that they have no conflicts of interest.

 Editor and Reviewer comments:    

First of all, we wish to thank the reviewers for the evaluation of our manuscript. We have revised the manuscript according to the anonymous reviewers’ comments. The comments and their replies are shown below in two different colors. We hope you will consider our manuscript for publication in your esteemed journal. The manuscript is original, no part of the manuscript has been published before, nor is any part of it under consideration for publication at another journal. We convey our immense thanks for dedicating your time to this evaluation. For, without doubt, it helped to improve the quality of this manuscript. Thank you.

REVIEWER 1

This paper delves into the application of space syntax to model pedestrian flows in both open and closed built environments on a global scale. Employing bibliographic methods and the PRISMA technique, the study meticulously analyzes relevant manuscripts, with a specific focus on the intricate concepts of pedestrian flows and their correlation to Sustainable Development Goal (SDG) 11. Emphasizing the paramount role of deliberate planning in urban design, the paper underscores the predictive capabilities of space syntax in crafting environments conducive to pedestrian movement. The conclusion accentuates the significance of space syntax in comprehending and enhancing global pedestrian flows, urging for its broader integration in forthcoming projects. Overall, the paper maintains a well-structured logical flow, ensuring ease of comprehension and aided by elucidative tables.

Authors respond: Thank you for your important words in acknowledging the quality of our manuscript. Thank you also for helping us to improve the quality of our manuscript.

While the cited literature aligns coherently with the paper's theme and exhibits adequacy in both quantity and quality, the authors can further fortify their work by considering insights from the following publication:

Authors respond: Thank you for your important words in acknowledging the quality of our manuscript. Thank you also for helping us to improve the quality of our manuscript.

Esposito, D., Santoro, S., & Camarda, D. (2020). "Agent-based Analysis of Urban Spaces Using Space Syntax and Spatial Cognition Approaches: A Case Study in Bari, Italy." Sustainability, 12(11), 4625.

This source offers a holistic understanding of how users experience and navigate urban spaces, providing valuable insights into the dynamics of user flows. The user-centric perspective presented in this paper contributes to a more comprehensive grasp of user flows in open and closed spaces, aligning seamlessly with the objectives of sustainable urban development.

Authors respond: Thanks for pointing this item out. As requested, the reference was inserted into the text of the manuscript. Thank you for helping us improve the quality of our manuscript. Thanks.

These considerations are expected to enhance the paper, offering a valuable perspective for further research on the topic and making it more suitable for publication.

Authors respond: Thank you for your important words in acknowledging the quality of our manuscript. Thank you also for helping us to improve the quality of our manuscript.

We point out that the text was submitted to a specialized professional who is a native English speaker, specializing in translating scientific manuscripts and to an agency specializing in spelling corrections to correct possible spelling and grammatical errors in the manuscript. Thank you for your important words in acknowledging the quality of our manuscript.

Thank you very much for taking the time to make these suggestions regarding the revision of our manuscript, thus markedly improving its organizational quality. Thanks.!

We are very grateful for the invaluable contributions by the reviewers, which further improved the quality of the manuscript. Technically, the only word of thanks we can express is sincere Gratitude.

Yours very sincerely,

___________________

Corresponding authors

Reviewer 2 Report

Comments and Suggestions for Authors

This article does not show that there is an understanding of the generic terms used in the field of urbanism, as well as little understanding of the field of Space syntax mentioned in the title of the article. Displacement for example is used instead of movement, ... Furthermore, the research question demonstrates a lack of basic understanding of what space syntax is. For example, the "theory of natural movement" specifically called as such is referred to in the paper as "the natural mobility of users", which is not what it is about. The meaning of terms such as "pedestrian deforestation", and many more, remains unclear.

The paper does not establish a clear rationale that links sustainability (SDG 11, Sustainable Cities and Communities) and space syntax, and why they should be approached through “a bibliographic study” (?) .

If it is a review of literature, as the use of the PRISMA method seems to suggest, the main journal articles dealing with natural movement and space syntax are missing, and the ones presented are too few (4) and not necessarily the most relevant.

Comments on the Quality of English Language

Too many unclear sentences for this paper to be understandable.

Author Response

Manuscript ID: architecture-2633507

Space Syntax at expression of science on user flows in open and closed spaces aimed at achieving SDGs: A review

 Dear Editor

Thank you for taking the time to manage this editorial process and for sending us review comments of revision.

We have revised our paper accommodating all the comments. We provide below the point by point reply to the comments.

We declare that this paper has not been published previously and it is not under consideration for publication elsewhere. The manuscript does not involve research on humans and animals. The authors also declare that they have no conflicts of interest.

 Editor and Reviewer comments:    

First of all, we wish to thank the reviewers for the evaluation of our manuscript. We have revised the manuscript according to the anonymous reviewers’ comments. The comments and their replies are shown below in two different colors. We hope you will consider our manuscript for publication in your esteemed journal. The manuscript is original, no part of the manuscript has been published before, nor is any part of it under consideration for publication at another journal. We convey our immense thanks for dedicating your time to this evaluation. For, without doubt, it helped to improve the quality of this manuscript. Thank you.

REVIEWER 2

This article does not show that there is an understanding of the generic terms used in the field of urbanism, as well as little understanding of the field of Space syntax mentioned in the title of the article. Displacement for example is used instead of movement, ... Furthermore, the research question demonstrates a lack of basic understanding of what space syntax is. For example, the "theory of natural movement" specifically called as such is referred to in the paper as "the natural mobility of users", which is not what it is about. The meaning of terms such as "pedestrian deforestation", and many more, remains unclear.

Authors respond: Thank you for highlighting this. Regarding terms, we do not agree with the statement that the article does not demonstrate an understanding of the generic terms used in the field of urbanism, and the other two reviewers also do not agree, as the authors are firmly rooted and published in the area of urbanism, some with high h-index ratings (https://www.scopus.com/authid/detail.uri?authorId=56973887600 / https://www.scopus.com/authid/detail.uri?authorId=55110642200). Therefore, we respectfully disagree with the argument. This manuscript works with practical issues, the term displacement was also used by other authors in the manuscripts reviewed. The term: “theory of natural movement" was adjusted in the text as recommended. We emphasize that the term "pedestrian deforestation" was an error, and was corrected in the text to “pedestrian displacement.” Thank you for your important words in acknowledging the quality of our manuscript. Thank you also for helping us to improve the quality of our manuscript.

The paper does not establish a clear rationale that links sustainability (SDG 11, Sustainable Cities and Communities) and space syntax, and why they should be approached through “a bibliographic study” (?).

Authors respond: Thank you for pointing this out. We make this relationship clear in the text by including a quote from: Esposito, D., Santoro, S., & Camarda, D. (2020). "Agent-based Analysis of Urban Spaces Using Space Syntax and Spatial Cognition Approaches: A Case Study in Bari, Italy." Sustainability, 12(11), 4625. Thank you for your important words in recognizing the quality of our manuscript. Thank you also for helping us to improve the quality of our manuscript.

If it is a review of literature, as the use of the PRISMA method seems to suggest, the main journal articles dealing with natural movement and space syntax are missing, and the ones presented are too few (4) and not necessarily the most relevant.

Authors respond: This manuscript indeed is a literature review article. Thus, we report that the use of the PRISMA method selected the main articles that deal with the application of space syntax in open and closed spaces; and not the discussion of the theory of natural motion and space syntax with application in theoretical studies. We agree that there are few, but that's what we found. We wish to remind you that 2,440 space syntax articles were analyzed to arrive at this number of selected articles, which deal with space syntax as applied to studies of open spaces and closed spaces on a global scale. Therefore, the importance of this article, and indeed our goal is to stimulate other studies that enhance the application of space syntax in open and closed spaces. Thank you also for helping us to improve the overall quality of our manuscript.

 Comments on the Quality of English Language: Too many unclear sentences for this paper to be understandable.

Authors respond: The authors point out that the text was submitted to a specialized professional who is a native English speaker, specializing in translating scientific manuscripts and to an agency specializing in spelling corrections to correct possible spelling and grammatical errors in the manuscript. Thank you for your important work in reviewing our article.

Thank you very much for taking the time to make these suggestions regarding the revision of our manuscript, thus markedly improving its organizational quality. Thanks.!

We are very grateful for the invaluable contributions by the reviewers, which further improved the quality of the manuscript. Technically, the only word of thanks we can express is sincere Gratitude.

Yours very sincerely,

___________________

Corresponding authors

Reviewer 3 Report

Comments and Suggestions for Authors

I would like to extend my thanks to the authors for their commendable effort in creating such a scientific work. This review paper is indeed noteworthy and has the potential for publication. However, to enhance its quality, several improvements are suggested based on the following comments.

1.       The paper's title should effectively convey its underlying idea to capture a broader audience. Therefore, I recommend replacing "user flow" with "people" to humanize it. Similarly, consider using "urban and architectural spaces" instead of "open and (close!) space."

2.       To make the abstract more informative, it would be beneficial if the authors clearly state the existing gaps they identified in the field.

3.       Clearly declare the association between SDG 11 and space syntax.

4.       Include "Space Syntax" in the keywords list.

5.       Figures should not be placed in the Introduction; consider relocating them.

6.       Revise the topic sentence of the introduction for greater clarity, ensuring it aligns with the paper's focus.

7.       The first paragraph seems somewhat irrelevant and off-topic; consider rephrasing it to provide a better explanation of the crucial variables in the study.

8.       The research is needed to be addressed by further relevant papers associated with social interactions, behavioral patterns, legibility, cognitive maps, sense of belonging, mobility patterns, spatial cognition, security, wayfinding, and privacy protection which used space syntax, to expand the applicability of space syntax in SDG 11. It is highly recommended to use the most recent documents published in 2020-2023.

9.       The authors missed an important section of ‘Inclusion’ in the PRISMA flowchart that need to be addressed.

10.   In the Abstract you mentioned to use both Scopus and WOS. However, in the PRISMA flowchart you mentioned that the authors used Scopus and Science Direct. The paper must be consistent and integrated and excluded from any sort of ambiguity. Revise it.

11.   Firstly, the authors need be clarified what do exactly they mean by open and close spaces. If architectural spaces included closed spaces, the number of manuscripts would be much higher than 4 items.

12.   There is a problem with citing references in page 4, line: 167.

13.   Please include the checklist of PRISMA in the form of a visualization to indicate that how the checklist applied in your review. Since your descriptions for methods is not thorough sufficiently, this figure may compensate this drawback.

14.   Reference number 23, 29, 30, 31, 32, 33, and 34 repeated multiple times within the manuscript that need to be refined.

15.   The authors are strongly suggested to add the sections of Results and Discussion to their paper in their relevant parts to ameliorate the structure of their paper.

16.   It would be so impressive, if the authors could provide a SANKEY diagram for their review paper to delineate the interrelation of Space Syntax applicability to the SDG 11 to reinforce their analytical approach.

17.   Clarifying the existing scientific gaps in the field is essential for the review paper. Recommendations and suggestions for orientations of future studies in this field is imperative.

18.   It would be more insightful if the authors highlight the novelty of their work in the conclusion.

Good luck

Author Response

Manuscript ID: architecture-2633507

Space Syntax at expression of science on user flows in open and closed spaces aimed at achieving SDGs: A review

 Dear Editor

Thank you for taking the time to manage this editorial process and for sending us review comments of revision.

We have revised our paper accommodating all the comments. We provide below the point by point reply to the comments.

We declare that this paper has not been published previously and it is not under consideration for publication elsewhere. The manuscript does not involve research on humans and animals. The authors also declare that they have no conflicts of interest.

 Editor and Reviewer comments:    

First of all, we wish to thank the reviewers for the evaluation of our manuscript. We have revised the manuscript according to the anonymous reviewers’ comments. The comments and their replies are shown below in two different colors. We hope you will consider our manuscript for publication in your esteemed journal. The manuscript is original, no part of the manuscript has been published before, nor is any part of it under consideration for publication at another journal. We convey our immense thanks for dedicating your time to this evaluation. For, without doubt, it helped to improve the quality of this manuscript. Thank you.

REVIEWER 3

I would like to extend my thanks to the authors for their commendable effort in creating such a scientific work. This review paper is indeed noteworthy and has the potential for publication. However, to enhance its quality, several improvements are suggested based on the following comments.

Authors respond: Thank you for your important words in acknowledging the quality of our manuscript. Thank you also for helping us to improve the quality of our manuscript.

  1. The paper's title should effectively convey its underlying idea to capture a broader audience. Therefore, I recommend replacing "user flow" with "people" to humanize it. Similarly, consider using "urban and architectural spaces" instead of "open and (close!) space."

Authors respond: Thank you for pointing out the item. The term users refers to those who use the space, according to the material covered in the literature, as there are people who do not use the circulation spaces, hence the importance of addressing the term users. In relation to open and closed terms, they are classic search terms used in this study. Thank you for helping us to improve the quality of our manuscript.

  1. To make the abstract more informative, it would be beneficial if the authors clearly state the existing gaps they identified in the field.

Authors respond: Thank you for pointing this out. This contextualization was inserted in the abstract of the manuscript, as suggested. Thank you for helping us to improve the quality of our manuscript.

  1. Clearly declare the association between SDG 11 and space syntax.

Authors respond: Thank you for highlighting this. This relationship was inserted in the abstract of the manuscript, as requested. Thank you so much for helping us to improve the quality of our manuscript.

  1. Include "Space Syntax" in the keywords list.

Authors respond: As suggested, the term Space Syntax was inserted in the keywords. Thanks.

  1. Figures should not be placed in the Introduction; consider relocating them.

Authors respond: Thank you for pointing this out. Following your request, the introduction figure was removed. Thank you for helping us to improve the written quality of our manuscript.

  1. Revise the topic sentence of the introduction for greater clarity, ensuring it aligns with the paper's focus.

Authors respond: Thank you. As requested, the introduction topics were revised. Thanks.

  1. The first paragraph seems somewhat irrelevant and off-topic; consider rephrasing it to provide a better explanation of the crucial variables in the study.

Authors respond: Thank you. Following your request, the first paragraph of the introduction was reformulated. Thanks.

  1. The research is needed to be addressed by further relevant papers associated with social interactions, behavioral patterns, legibility, cognitive maps, sense of belonging, mobility patterns, spatial cognition, security, wayfinding, and privacy protection which used space syntax, to expand the applicability of space syntax in SDG 11. It is highly recommended to use the most recent documents published in 2020-2023.

Authors respond: Thank you for highlighting this. Valuing your comments, we included your recommendations in the conclusion as a recommendation for future studies. This article addressed practical studies of open spaces and closed spaces within the scope of space syntax, as related to sustainability on a global scale, as selected from 2,440 articles. Therefore, the importance of this article is to stimulate other studies that enhance the application of space syntax in open spaces and closed spaces, demonstrating the little practical application of space syntax that deal with open spaces and closed spaces in an applied way. Thank you sincerely for helping us to improve the overall quality of our manuscript.

  1. The authors missed an important section of ‘Inclusion’ in the PRISMA flowchart that need to be addressed.

Authors respond: Thank you for pointing this out. To remedy this issue, more details were inserted in the figure. Thank you for helping us to improve the written quality of our manuscript.

  1. In the Abstract you mentioned to use both Scopus and WOS. However, in the PRISMA flowchart you mentioned that the authors used Scopus and Science Direct. The paper must be consistent and integrated and excluded from any sort of ambiguity. Revise it.

Authors respond: Thank you for pointing out the item. The right item (Scopus and ScienceDirect) was corrected in the text. Thank you for helping us to improve the quality of our manuscript.

  1. Firstly, the authors need be clarified what do exactly they mean by open and close spaces. If architectural spaces included closed spaces, the number of manuscripts would be much higher than 4 items.

Authors respond: Thank you. In fact, this study is not related to architectural spaces, it is related to the applications of space syntax in open spaces and closed spaces. An application of architectural spaces would be another study. Thank you.

  1. There is a problem with citing references in page 4, line: 167.

Authors respond: Thank you for highlighting this. As recommended, the citation was corrected in the text. Thank you for helping us to improve the quality of our manuscript. Thanks.

  1. Please include the checklist of PRISMA in the form of a visualization to indicate that how the checklist applied in your review. Since your descriptions for methods is not thorough sufficiently, this figure may compensate this drawback.

Authors respond: Thank you for this suggestion. As recommended, the representation of the Prisma was better detailed. Thank you for helping us to improve the written quality of our manuscript.

  1. Reference number 23, 29, 30, 31, 32, 33, and 34 repeated multiple times within the manuscript that need to be refined.

Authors respond: Thank you. As recommended, all references were reviewed, and each that remain deal directly with what is written in the section of each quote. Thank you for helping us to improve the overall quality of our manuscript.

  1. The authors are strongly suggested to add the sections of Results and Discussion to their paper in their relevant parts to ameliorate the structure of their paper.

Authors respond: As recommended, the title Results and Discussion was inserted. Thank you for helping us to improve the quality of our manuscript. Thanks.

  1. It would be so impressive, if the authors could provide a SANKEY diagram for their review paper to delineate the interrelation of Space Syntax applicability to the SDG 11 to reinforce their analytical approach.

Authors respond: Thank you once again. As recommended, a SANKEY diagram was inserted into the manuscript. Thank you for helping us to improve the overall quality of our manuscript.

  1. Clarifying the existing scientific gaps in the field is essential for the review paper. Recommendations and suggestions for orientations of future studies in this field is imperative.

Authors respond: Thank you. As recommended, this was revised in the text. Thank you for helping us to improve the quality of our manuscript. Thanks.

  1. It would be more insightful if the authors highlight the novelty of their work in the conclusion. Good luck

Authors respond: Thank you for suggesting this item. As recommended, this was included in the conclusion. Thank you for helping us to improve the overall quality of our manuscript.

We point out that the text was submitted to a specialized professional who is a native English speaker, specializing in translating scientific manuscripts and to an agency specializing in spelling corrections to correct possible spelling and grammatical errors in the manuscript. Thank you for your important words in acknowledging the quality of our manuscript.

Thank you very much for taking the time to make these suggestions regarding the revision of our manuscript, thus markedly improving its organizational quality. Thanks.!

We are very grateful for the invaluable contributions by the reviewers, which further improved the quality of the manuscript. Technically, the only word of thanks we can express is sincere Gratitude.

Yours very sincerely,

___________________

Corresponding authors

Round 2

Reviewer 3 Report

Comments and Suggestions for Authors

I would like to thank the authors for adopted revisions.

I am pleased to let you know that your paper is acceptable now.

Good luck and congratulations